



# Cropland expansion drives vegetation greenness decline in Southeast Asia

Ruiying Zhao[1*], Xiangzhong Luo[1, 2*], Yuheng Yang[1], Luri Nurlaila Syahid[1], Chi Chen[3], Janice Ser Huay Lee[4]

[1]Department of Geography, National University of Singapore, 117570, Singapore
[2]Center for Nature-based Climate Solutions, Department of Biological Sciences, National University of Singapore, 117558, Singapore
[3]Department of Ecology, Evolution, and Natural Resources, School of Environmental and Biological Sciences, Rutgers University, New Brunswick, NJ 08901, USA
[4]Asian School of the Environment and Earth Observatory of Singapore, Nanyang Technological University, 637459, Singapore

*Correspondence to*: Ruiying Zhao (ruiying@nus.edu.sg); Xiangzhong Luo (xzluo.remi@nus.edu.sg)

**Abstract.** Land use and land cover changes (LUCC) is a key factor in determining regional vegetation greenness, impacting terrestrial carbon, water, and energy budgets. As a global hotspot of LUCC, Southeast Asia has experienced intensive cropland

and plantation expansions in the past half-century, yet their impacts on regional greenness have not been elucidated. Here, we harmonized multiple land cover datasets, and used satellite-derived leaf area index (LAI) in combination with a machine learning approach to quantify the impacts of LUCC on vegetation greenness in insular Southeast Asia (i.e., Peninsular Malaysia, Sumatra, and Borneo islands). We found that regional LAI shows almost no trend ($0.04 \times 10^{-2}$ m$^2$ m$^{-2}$ yr$^{-1}$) from 2000 to 2016, as a net effect of increased LAI ($+5.71 \times 10^{-2}$ m$^2$ m$^{-2}$ yr$^{-1}$) due to CO2 fertilization, offset by decreased LAI

mainly due to cropland expansion ($-4.46 \times 10^{-2}$ m$^2$ m$^{-2}$ yr$^{-1}$). The impact of croplands on greenness in Southeast Asia contrasts with that in India and China. Meanwhile, oil palm expansion and climate change induced only small decreases in LAI in Southeast Asia ($-0.41 \times 10^{-2}$ m$^2$ m$^{-2}$ yr$^{-1}$ and $-0.38 \times 10^{-2}$ m$^2$ m$^{-2}$ yr$^{-1}$, respectively). Our research unveils how LAI changes with different processes of LUCC in Southeast Asia and offers a quantitative framework to assess vegetation greenness under different land use scenarios.

## 1 Introduction

Terrestrial vegetation plays a pivotal role in regulating ecosystem services, conserving biodiversity, and mitigating climate change impacts. Over recent decades, long-term satellite records of the leaf area index (LAI) have disclosed a notable increase in vegetation greenness on Earth (Chen et al., 2019; Zhu et al., 2016). While at the global scale, elevated atmospheric CO$_2$ concentrations and climate change are regarded as the driving factors for vegetation greening (Zhu et al., 2016), at the regional

scale, land use and land cover changes (LUCC) can also substantially impact greenness. Previous studies have found that cropland intensification and afforestation are the primary drivers for the greening in India and China (Chen et al., 2019;



Kuttippurath and Kashyap, 2023). Meanwhile, other studies reported that deforestation for croplands or pastures serves as a key driver for decreasing greenness in Amazon (Chen et al., 2019; Querino et al., 2016). Thus, the LUCC can either increase or decrease vegetation greenness, depending on the prior land use (e.g., forests, pastures), the subsequent land use (e.g.,

croplands, pastures, and plantations), and the intensity of these land uses (Wang and Friedl, 2019).

Southeast Asia has been a global hotspot of LUCC since the 1950s (Houghton and Nassikas, 2017), with maritime countries such as Indonesia and Malaysia having the greatest deforestation rates in the world (Harris et al., 2012). An increase in food crops and export-oriented crop production has driven a significant transformation of tropical forests into commodity plantations like oil palms or croplands for food (Fagan et al., 2022; Zeng et al., 2018). Indonesia and Malaysia are the largest

producers of palm fruit, with 250 and 97 million tons of palm fruit produced in 2020 (FAOSTAT, 2022), respectively. Meanwhile, Indonesia and Malaysia experienced tree cover losses of approximately 29.4 and 8.92 million hectares respectively in the past two decades, equivalent to 18% and 30% of their tree cover in 2000 (Global Forest Watch, https://www.globalforestwatch.org/).

Despite the substantial LUCC in the past years, we lack a clear understanding of the impacts of LUCC on vegetation greenness

in Southeast Asia. This is partly due to the complexity of the recent land use history of the region. For example, in Indonesia, the dominant LUCC types in the 2000s were the conversion of forests to oil palm plantations and cropland in lowland regions, while in the 2010s, the conversion of forests to oil palm slowed down, shifting more towards highland croplands and the rotation of plantations (Descals et al., 2021; Xu et al., 2020; Zeng et al., 2018). These LUCC processes can differentially affect vegetation greenness and biogeochemical cycles (Ito and Hajima, 2020), with further feedbacks on vegetation greenness

(Wang and Friedl, 2019). However, current studies on the assessment of LUCC impacts do not often distinguish these individual land use processes, but instead categorize the loss of forests under one "deforestation" category (Sitch et al., 2015) or regard plantation as similar to natural forests (Hansen et al., 2013).

In this study, we aim to assess the impact of LUCC on vegetation greenness in Southeast Asia and quantify the contributions of the different LUCC processes to the changes in greenness. We collected and harmonized various types of land cover datasets

(Chini et al., 2021; Hansen et al., 2013; Sulla-Menashe et al., 2019; Xu et al., 2020) to build a detailed land use history for Southeast Asia from 2000 to 2016. We further used a machine learning approach to quantify the impacts of land uses on LAI, along with the impacts from climate, $CO_2$ concentrations, stand age, etc. Our machine learning approach, combined with hypothetical scenarios that simulate vegetation greenness without LUCC processes, enables us to isolate the impacts of different land use changes on LAI by estimating the difference between scenario-based LAI.

## 2. Materials and Methods

### 2.1 Study areas

Depending on the availability of various land use data, we focused our study on a region including Peninsular Malaysia, Sumatra, and Borneo islands (Fig. 1), which experience rapid LUCC in Southeast Asia (Geist and Lambin, 2001; Mao et al.,



2023). Since the 1980s, the insular Southeast Asia has lost at least 1.0% of its forests annually (Felbab-Brown, 2013; Miettinen

et al., 2011), primarily due to cropland and plantation expansions (Wang et al., 2023; Wicke et al., 2011; Xu et al., 2020). Particularly notable is the expansion of oil palm plantations in Indonesia and Malaysia. From the 1990s to the 2010s, the extent of oil palm plantations increased from 1.3 million hectares (Mha) to 7.7 Mha in Indonesia and from 2.1 to 5.2 Mha in Malaysia (Xu et al., 2022). A national-wide study reported that around 55% to 59% of oil palm plantation expansions in Malaysia and at least 56% in Indonesia occurred on lands previously covered by forests during the period of 1990 to 2005 (Koh and Wilcove,

2008; Vijay et al., 2016). In parallel, cropland expansion also drives deforestation in our study area, with approximately 15% of forest loss in Indonesia attributed to this cause (Austin et al., 2019). Rubber, timber and other plantations have also resulted in deforestation. A recent study revealed that between 2001 and 2016, approximately 20% of rubber plantations in Indonesia and 33% in Malaysia were established on land previously covered by forests, resulting in a loss of about 1 Mha and 0.32 Mha of forest in these countries, respectively (Wang et al., 2023).

**2.2 Identification of greening trend**

Leaf area index indicates the total amount of one-sided leaf area per unit ground surface area (Chen and Black, 1992; Watson, 1947) and often serves as a measure of vegetation greenness (Zhu et al., 2016). In this study, we use the GLOBMAP LAI dataset, which provides a global record of vegetation cover with a 500 m resolution and is one of the main LAI datasets used for global greenness studies (Piao et al., 2020; Zhu et al., 2016). We used the GLOBMAP LAI dataset post 2001, which was

generated based on the Moderate Resolution Imaging Spectroradiometer (MODIS) surface reflectance (Liu et al., 2012), with an advanced algorithm to consider canopy clumping, making it particularly suitable for dense canopies in the tropics (Fang et al., 2019). To assess the trend of LAI for individual pixels, we utilized the Mann-Kendall Test, a non-parametric statistical method that can effectively identify consistent upward or downward trends over time (Mann, 1945). This test provides the pixel-by-pixel magnitude ($\beta$) and statistical significance ($p$-values) of the greening trends.

**2.3 Mapping different land-use transitions**

In our study area, we considered natural forests, oil palms, and croplands as the major land use types, as they together accounted for over 90% of the land cover. To delineate the annual changes of all these land use types from 2001 to 2016, we harmonized various land use datasets, which were provided at different spatial resolutions (Table S1), into a unified dataset on a 500 m grid. This harmonization was essential to make changes in various land use types compatible with each other, and to match

the spatial resolution of the land use data to LAI dataset. As Fig. S1 shows, the workflow proceeded as follows: (1) we first determined the annual percentage of forested (A%) and non-forested areas (B%) within each 500m grid cell, by aggregating the mean of the annual 30 m resolution Global Forest Change (GFC) maps (Hansen et al., 2013); (2) Within the fraction of forested area in each grid cell, we estimated the proportion of oil palm (OP) plantations (A1%) based on an openly available dataset for oil palm distribution spanning from 2001 to 2016 across Malaysia and Indonesia (Xu et al., 2020). The dataset

provides an annual OP distribution at 100 m resolution, generated using observations from Advanced PALSAR, ALOS-2



PALSAR-2, and MODIS. To estimate the proportion of OP, we calculated the frequency of oil palm pixels in each 500 m × 500 m window. (3) After accounting for the area of OP, the remaining forested area in each grid was further categorized into the evergreen broadleaf forest (EBF) (A2%) and other forest types (i.e., deciduous broadleaf forest, coniferous forest, mixed forest, etc.), based on the ratio of EBF to the total forested area provided by MODIS Land Cover Type Product (MCD12Q1);

(3) Within the non-forested fraction of each grid cell, we used a recent version of the Land-use harmonization datasets (LUH2) dataset to estimate the percentage of cropland (CRO) (B1%) and other non-forest vegetated land uses (i.e., pasture, grass, etc.). In this analysis, we assumed that the fraction of each land use type in the LUH2 dataset on a 0.25° grid is applicable to the 500 m grid cells within each 0.25° grid cell. At the end, we obtained detailed information for EBF, OP, CRO, and "Other" land-use types (including other forests and non-forest vegetated areas), at the 500 m spatial resolution. We grouped other forests

and other non-forest vegetated areas together, as they represented a minor proportion (less than 5%) of the land surface (Table S2) and exhibited minimal changes during the study period.

## 2.4 Extreme gradient boosting model

Tree-based machine learning models, such as Extreme Gradient Boosting (XGBoost), have been widely used in predicting and analysing ecosystem dynamics (Green et al., 2022; Wang et al., 2022a; Yuan et al., 2019). Compared to neural networks,

which often function like 'black boxes', tree-based models offer greater interpretability and are particularly effective on tabular-style datasets (Lundberg et al., 2020). XGBoost is an ensemble learning algorithm that iteratively constructs multiple decision trees and has proven to be effective for both classification and regression tasks (Chen and Guestrin, 2016; Yan et al., 2020). This algorithm employs shrinkage techniques and performs multithreaded calculations to minimize overfitting (Meng et al., 2021).

In our study, we applied the XGBoost algorithm (Chen and Guestrin, 2016) to model the spatial-temporal variations in the mean annual LAI using climatic and LUCC factors as inputs. These factors include the fractions of EBF, OP, CRO, and Other land uses, EBF-stand and OP-stand ages (Besnard et al., 2021; Danylo et al., 2021), $CO_2$ concentrations (https://gml.noaa.gov/ccgg/trends/), and climatic variables (Table S3). The climatic variables in our study include mean annual temperature (MAT), mean annual precipitation (MAP), wind speed (WS), shortwave radiation (RAD) and relative humidity

(RH). These gridded climatic variables were obtained from the European Centre of Medium-Range Weather Forecasts (ECMWF) reanalysis product v5 - Land (https://cds.climate.copernicus.eu/cdsapp#!/dataset/reanalysis-era5-land). We aggregate the original hourly data to the annual time step using annual average.

To fine-tune the parameters of our XGBoost model for LAI prediction, we utilized the GridSearchCV method to test different parameter combinations (i.e., varying numbers of trees from 150 to 400, tree depths from 5 to 15, and learning rates between

0.01 and 0.1) and determined the best parameter combinations through cross-validation (Pedregosa et al., 2011a).




## 2.5 Shapley Additive Explanations

We utilized the TreeExplainer-based SHapley Additive exPlanations (SHAP) framework to interpret the individual and interactive contributions of LUCC and other factors (including climate variables, $CO_2$ concentration, and stand ages) to the LAI variations in our XGBoost model. The SHAP methodology, which is based on the concept of Shapley values in cooperative game theory, offers an insightful interpretation of factor importance (Lundberg and Lee, 2017a). It provides detailed, instance-specific explanations, termed SHAP values to quantify the impact of each factor on the model predictions (Lundberg et al., 2020; Yang et al., 2021).

In the SHAP framework, the value for a given factor $i$ in a particular sample $x$ is computed as the average marginal contribution of that factor across all possible combinations. This is mathematically represented as:

$$\phi_i(x) = \sum_{S \subseteq N\{i\}} \frac{|S|!\,(|N| - |S| - 1)!}{|N|!} [f(S \cup \{i\}) - f(S)] \tag{1}$$

where $N$ is the set of all factors, $S$ is a subset of factors, and $f$ is the prediction model. This formula quantifies the contribution of factor $i$ by comparing the prediction with and without the factor, averaging over all possible subsets of factors.

The SHAP value indicates the magnitude and direction of the impact of a factor on prediction in the specific sample. To be specific, the magnitude (absolute value) of a SHAP value indicates the importance of a factor. Larger absolute SHAP values mean the factor has a greater impact on the model's output. The sign of a SHAP value (positive or negative) shows the direction of the impact. A positive SHAP value indicates that the factor positively affects the model's output (e.g., increases LAI), while a negative SHAP value suggests a negative impact (e.g., decreases LAI). By aggregating mean SHAP values of all samples, we can also derive global factor importance, which offers a holistic view of variables affecting annual LAI variations.

Furthermore, SHAP values aid in the interpretation of the interactive effects of two or more factors in machine learning models. The interactive effect is defined as the change in prediction when the joint contribution of two or more factors is considered, by subtracting the individual contributions made by each factor. The interactive effects of $i^{\text{th}}$ and $j^{\text{th}}$ factors are expressed as,

$$\phi_i, j(x) = \sum_{S \subseteq N \setminus \{i\}} \frac{|S|!\,(|N| - |S| - 2)!}{|N|!} [f(S \cup \{i,j\}) - f(S \cup \{i\}) - f(S \cup \{j\}) + f(S)] \tag{2}$$

We utilized the *xgboost* and *scikit-learn* packages in Python 3.11.0 for developing and training the XGBoost model (Chen and Guestrin, 2016; Pedregosa et al., 2011b). Then, we employed the 'TreeExplainer' function from the *shap* (Lundberg and Lee, 2017b) package to interpret the impact of factors on LAI predictions.

## 2.6 Simulation scenarios

To quantify and compare the impacts of specific LUCC processes, climate change, and elevated $CO_2$ concentrations on vegetation greenness changes, we estimated the LAI trend under various hypothetical scenarios using the established XGBoost model. We used 5 scenarios (S1, S2, S3, S4, S5) in the study:



S1: $CO_2$ only (i.e., time-varying $CO_2$ from 2001 to 2016), with climate and land uses remaining unchanged (i.e. using the values in 2001). This scenario simulated the LAI trend under the impact of elevated $CO_2$. S2: $CO_2$ and climate change (i.e., time-varying $CO_2$ and climate from 2001 to 2016), with no changes in land uses. We then quantified the impact of climate change based on the difference in LAI trends between S2 and S1.

S3 to S5 sequentially considered different land use processes. S3: EBF to CRO using time-varying $CO_2$, climate change, and
CRO area, while keeping the area of OP and other land use types remained unchanged after 2001; S4: EBF to both CRO and OP using time-varying $CO_2$, climate change, CRO and OP areas, with other land uses unchanged since 2001; S5: EBF to all LUCC where all variables including $CO_2$, climate, and all types of LUCC are time-varying. We then quantified the impacts of each type of LUCC on vegetation greening based on differences in LAI trends between scenarios.

## 3. Results

### 3.1 Land Use Changes and Greening Trends

From 2001 to 2016, the extent of forest in our study area decreased annually by 1.29%, leading to a reduction in EBF from 73.41% to 53.09% of the study area. Notably, approximately 25.54% of the region experienced rapid deforestation with a forest loss rate exceeding 2% of the land surface per year. This deforestation rate was especially pronounced in the eastern parts of Sumatra and along the western and southern edges of Borneo (Fig. 1b).

In the deforested areas, we observed widespread expansions of CRO and OP plantations. The area of CRO increased at a rate of 0.63% per year, resulting in an increase of CRO from 14.45% to 24.56% of the region (i.e., $20.76 \times 10^4$ km$^2$ to $35.28 \times 10^4$ km$^2$) from 2000 to 2016. Meanwhile, the expansion of OP proceeded at a pace of 0.48% per year, resulting in nearly tripled the extent of oil plantations over the past decade (i.e., from 3.91% in 2001 to 12.05% in 2016; Fig. 1a). In central Sumatra, the south edge of Borneo, and the southern part of Peninsula Malaysia, OP showed the largest increase, partly at the expense of
decreases in CRO (Fig. 1c).



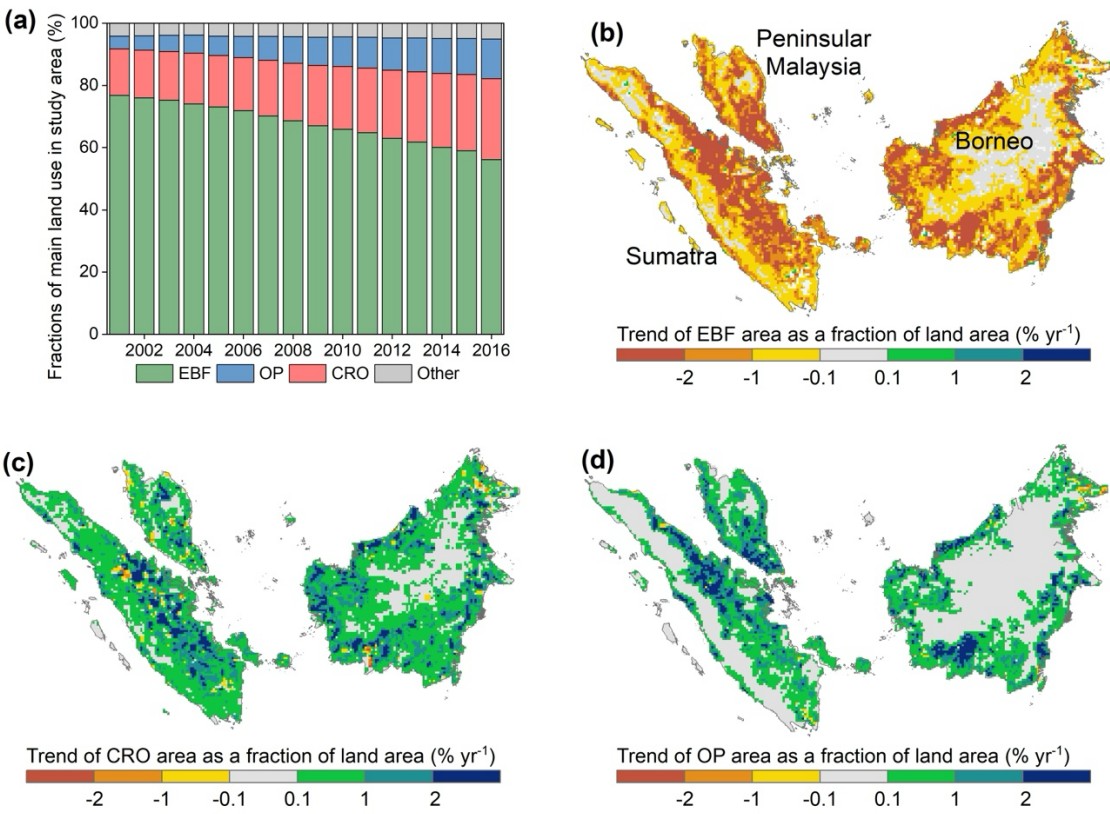

**Figure 1: Land use composition and its changes from 2001 to 2016 in the study area. (a) The changes of the fractions of main land uses including evergreen broadleaf forest (EBF), oil palm (OP), cropland (CRO), and others over the study period. (b)-(d) Spatial distribution of the trend of EBF, CRO, and OP as a fraction of land area.**

The regional average LAI exhibited a non-significant upward trend over the study period, with a slope of $0.04 \times 10^{-2}$ m$^2$ m$^{-2}$ yr$^{-1}$ (Fig. 2a). According to (Galán-Acedo et al., 2021), areas with over 70% forest loss are categorized as having high to severe deforestation, while areas with less than 70% forest loss are classified as undergoing low to intermediate deforestation. We found that for our study area, there was a non-significant upward trend in LAI. This was due to a net effect of a rapid LAI increase ($\beta = +1.07 \times 10^{-2}$ m$^2$ m$^{-2}$ yr$^{-1}$, $p < 0.05$) in areas with low to intermediate deforestation and a significant LAI decline

($\beta = -3.08 \times 10^{-2}$ m$^2$ m$^{-2}$ yr$^{-1}$, $p < 0.001$) in areas with high to severe deforestation (Fig. 2a). Across the study area, 58.50% of the region showed a significant decreasing LAI trend and they are mostly areas with pronounced forest loss (Fig. 1b).



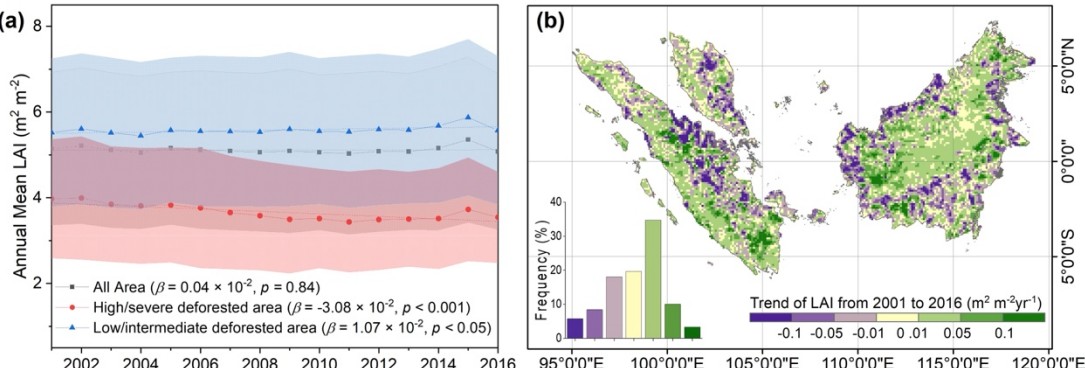

**Figure 2: The trends of LAI from 2001 to 2016 in the study area. (a) Regional average LAI trend, the LAI trends from regions with high to severe deforestation, and LAI trend from regions with low to intermediate deforestation. The classification of the deforestation level is referred to (Galán-Acedo et al., 2021), where areas experiencing more than 70% forest loss are classified as high/severe deforestation, whereas those with less than 70% loss are classified as low/intermediate deforestation. (b) the spatial pattern of LAI trend, with histogram plot shows the frequency (%) of the pixel-wise LAI trend in the study area.**

## 3.2 Drivers of the changes in LAI

To understand the variations in vegetation greenness in Southeast Asia, we established a XGBoost model to quantify the relationship between vegetation greenness and land uses, climate variables, $CO_2$ concentrations and stand ages. The XGBoost model showed high explanatory power (i.e., 98% accuracy for calibration and 93% accuracy for validation), underscoring the model's reliability for analysing the determinants of LAI variability (Fig. 3, Fig. S3 and S4).

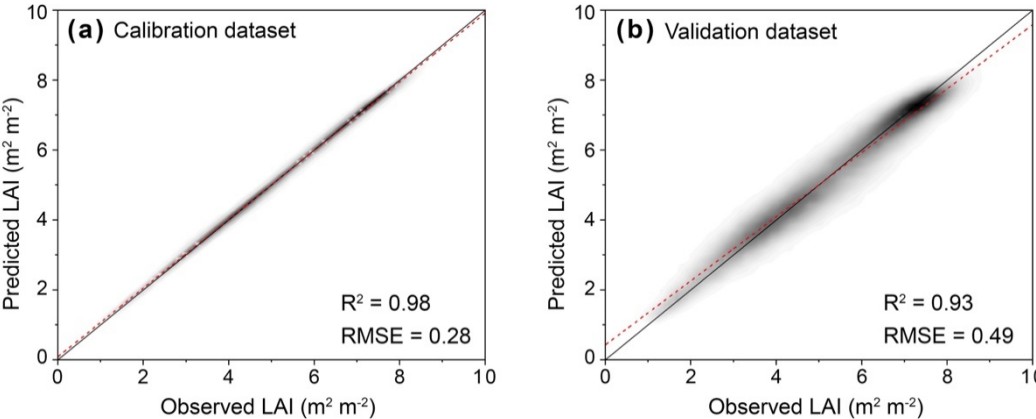

**Figure 3: Comparison of observed and predicted LAI values with the XGBoost model for the calibration dataset (a), and validation dataset (b).**





Based on the built XGBoost model, we evaluated the contributions of each factor to LAI in the SHAP framework (Fig. 4; see Methods). Our results revealed that the fraction of EBF (f_EBF) in each grid holds the greatest mean absolute SHAP value (1.28), indicating that f_EBF had the largest impact on LAI (Fig. 4b) and this impact was positive (Fig. 4a and Fig. S2a). The average impact of the fraction of OP (f_OP) on LAI ranked the second largest (0.21) and had a similar positive impact on

LAI as f_EBF (Fig. 4a, b and S2b). In contrast, the impact of CRO and other land uses on LAI was found to be negative (Fig. 4a, S2c and S2d). A higher fraction of CRO (f_CRO) led to a larger negative SHAP value. We further explored the impacts of interactions of land use types (i.e., f_CRO, f_OP and f_EBF) on LAI. The results demonstrated that in areas with low f_EBF, an increase of f_OP enhanced LAI while f_CRO induced LAI decreases. Meanwhile, in areas with high f_EBF, the impacts of both f_OP and f_CRO on LAI are markedly reduced, suggesting low intensity of land use change in the region (Fig. 4c and

4d).

Apart from the impacts of LUCC on LAI, we also found that elevated $CO_2$ concentration substantially increased LAI, with a mean SHAP value at 0.18. In contrast to elevated $CO_2$ concentration, the MAT was negatively related to LAI with a smaller mean SHAP value of 0.08 (Fig. 4b and S2e). Other climate variables have limited impacts on LAI. It is noteworthy is that the stand ages of both EBF (Age_EBF) and stand ages of OP (Age_OP) positively impact LAI. Specifically, Age_EBF has a

greater impact than that of Age_OP, as indicated by a higher absolute mean SHAP value for Age_EBF (0.11) compared to Age_OP (0.08). While we found that Age_EBF continuously contributed to LAI increases (Fig. S2k), Age_OP increased LAI at a younger age (less than 12 years) and then decreased LAI afterwards (Fig. S2l).





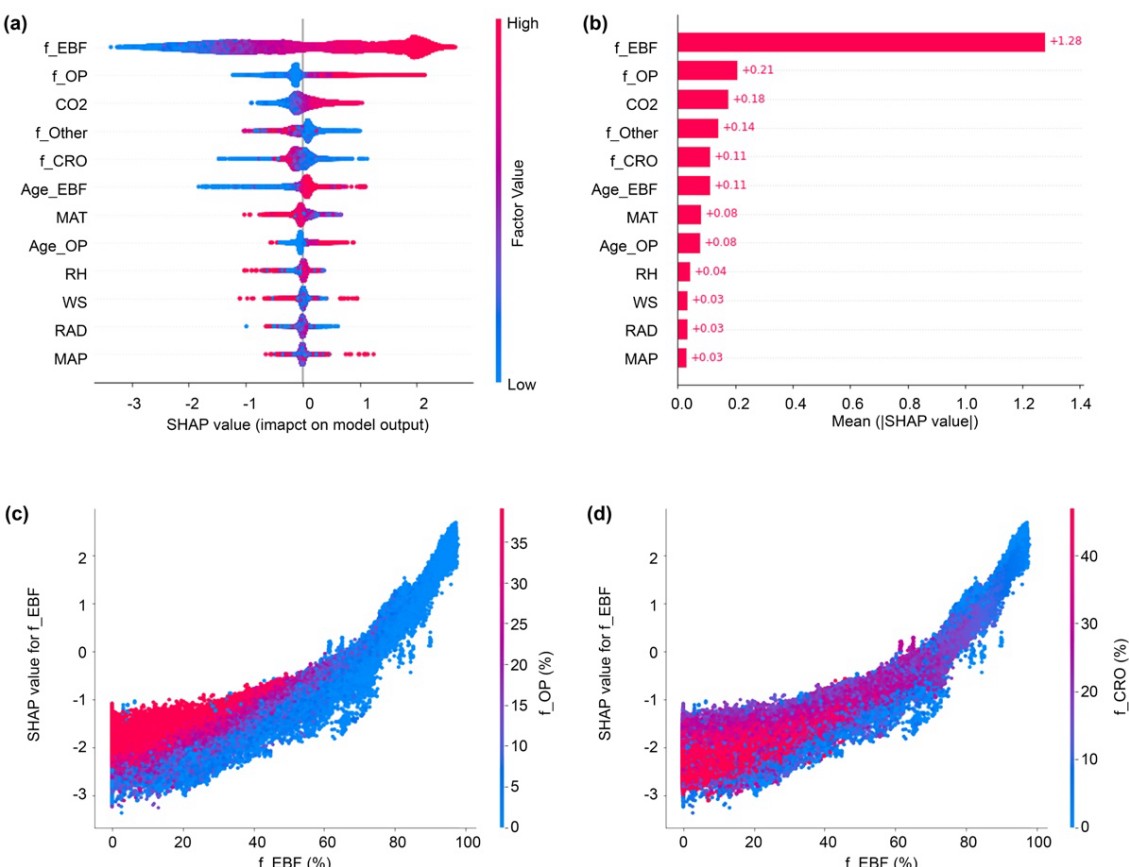

**Figure 4: The impact of factors on LAI. (a) Bee swarm plots show the SHAP values of each factor on LAI for each sample. The** 220 **SHAP value indicates the magnitude and direction of the impact on LAI (see Methods). Each dot represents an individual sample, with the color indicating the relative values of the specific factor. (b) The bar plot of the mean absolute SHAP values of each factor for LAI. (c) The interaction of f_OP and f_EBF, and (d) the interaction of f_CRO and f_EBF on LAI. The abbreviations for each factor are available in Table S3.**

### 3.3 Impacts of LUCC on Greening

Through scenario-based prediction, we quantified the impacts of LUCC, elevated $CO_2$ concentration and climate change on the trend of greening (Fig. 5f). Compared to the observed greening trend for the study area (i.e., $0.04 \times 10^{-2}$ $m^2$ $m^{-2}$ $yr^{-1}$), we found that the greening trend increased to $5.71 \times 10^{-2}$ $m^2$ $m^{-2}$ $yr^{-1}$ under the scenario S1, which simulated the effect of elevated $CO_2$ alone with climate and LUCC remaining constant. The result suggests that elevated $CO_2$ was the primary reason for the increase of LAI in Southeast Asia if there were. Climate change showed a small negative impact (i.e., mostly due to rising 230 temperature) on the LAI. We found both CRO expansion and OP expansion decreased LAI trend, with the trend dramatically dropped by $-4.46 \times 10^{-2}$ $m^2$ $m^{-2}$ $yr^{-1}$ under the impact of CRO expansion, and by $-0.41 \times 10^{-2}$ $m^2$ $m^{-2}$ $yr^{-1}$ under the impact of OP expansion. The results highlight that CRO expansion was the primary reason for the decrease in vegetation greenness,




counteracting the greening trend caused by elevated $CO_2$ in our study area. In contrast, OP expansion only contributed to a small decline in greenness.

We further examined the spatial variations of the impacts of each factor on greening by quantifying the differences in greening trend under different scenarios at the pixel level. Across the study area, LUCC imposed a negative impact on LAI trends (Fig. 5c, 5d and 5e). Consistent with regional average values, we found the changes from EBF to CRO had a more pronounced negative impacts on the greening trend than the conversion of EBF to OP (Fig. 5c and 5d). In some regions, such as the southern edges of Sumatra and Borneo, OP enhanced regional greening (Fig. 5d). Meanwhile, elevated $CO_2$ concentrations consistently

had a positive impact on greening across the region (Fig. 5a), and climate change showed an overall negative but highly heterogeneous impacts on LAI trends (Fig. 5b).

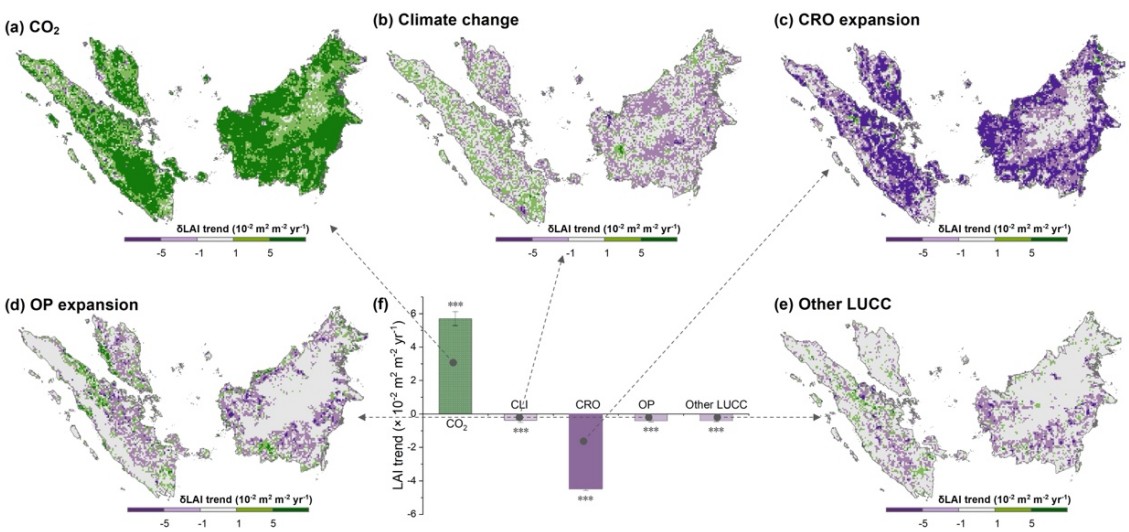

**Figure 5: The spatial distribution of the pixel-wise impacts of each factor on the greening trends (δLAI trend). Positive values mean the factors considered increase LAI trend and negative values mean otherwise. We show the spatial patterns of contribution from**
**(a) elevated $CO_2$ concentrations, (b) climate changes, (c) expansion of cropland (CRO), (d) expansion of oil palm (OP), and (e) other land use changes. (f) shows the average impact of each factor, with the error bars indicating one standard deviation. The symbol (∗∗∗) denotes a statistically significant difference in LAI trends at $p < 0.001$ level.**

## 4. Discussion

In this study, we analyzed the typical LUCC processes in Peninsular Malaysia, Sumatra and Borneo and their impacts on
greenness over the past two decades. We found a significant decline in EBF coverage, from 73.41% to 53.09%, predominantly due to CRO and OP plantation expansions. Meanwhile, we did not find a significant trend in LAI in our study area, as the increases in regional greenness due to elevated $CO_2$ were offset by the decreases in regional greenness caused by CRO and OP expansions. Notably, the negative effect of CRO expansion (-4.46 × $10^{-2}$ $m^2$ $m^{-2}$ $yr^{-1}$) was more pronounced than that of OP expansion (-0.41 × $10^{-2}$ $m^2$ $m^{-2}$ $yr^{-1}$), indicating a dominant role of CRO expansion on greenness decline in our study area.



## 4.1 Strong negative impact of cropland expansion on greenness in Southeast Asia

Our results demonstrate that CRO expansion in Southeast Asia contributed negatively to vegetation greenness. This is opposite to reports on the greening trend in China and India, where CRO are suggested as the main reason for the net increase in LAI (Chen et al., 2019; Kuttippurath and Kashyap, 2023). This disparity may be partly attributed to original land use types before CRO expansion. In Southeast Asia, CRO expansion mainly occurs at the expense of natural forests (Potapov et al., 2022; Zeng et al., 2018), and crops often have less dense canopies than the natural forests (Asner et al., 2005; Foley et al., 2005; Pocock et al., 2010). In contrast, the increase in LAI in China and India primarily resulted from the intensification of croplands, rather than their expansion, and where expansion did occur, it predominantly took place on lands that were previously bare or sparsely vegetated (Chen et al., 2019).

We also note that agricultural practices in Indonesia and Malaysia are generally less intensive compared to India and China, partly due to less advanced agricultural technologies deployed in the region (Liu et al., 2021). For example, in China and India, intensive agricultural practices including precision fertilization and advanced irrigation systems, such as drip irrigation in China and spray irrigation in India, are widely adopted to enhance crop growth (Wang et al., 2013; Cui et al., 2022). In addition, the development of specialized crop varieties, such as hybrid rice in China and climate-smart drought-tolerant rice varieties in India (Panda et al., 2021; Zhang et al., 2022), also facilitated plant growth and, consequently, regional greening (Zhao et al., 2021; Zhao et al., 2023). In contrast, Indonesia and Malaysia predominantly depend on rainfed irrigation and traditional farming methods, such as the Subak terraced rice fields in Indonesia, leading to lower crop intensity. Less intensive practices are less likely to create dense canopy and biomass of croplands (Takeshima, 2019).

## 4.2 Minor contribution of oil palm to regional greenness

Our study observed a negative, yet small impact of OP expansion on greenness. This aligns with studies reporting that OP has LAI values comparable to, or only slightly lower than, native forests (Propastin, 2009; Rusli and Majid, 2014; Vernimmen et al., 2007). Therefore, OP expansion, though a major driver for land use change in Southeast Asia, was not a main driver of greenness decline in the region.

Meanwhile, we also found that the age of OP stands influences LAI, as the LAI of OP increases stand age when OP is young, and then decrease LAI after a threshold of stand age. This observation agrees with previous studies reporting the plateau and decreases of oil palm LAI after the age of 10 (Xu et al., 2021), and the plateau and decrease of oil palm yield after age of 8-9 (Park et al., 2023). The stand age of most OP planted in Southeast Asia is approaching maturity (over half are planted before 2009) (Danylo et al., 2021). In response, industrial and smallholder plantations have undergone or are starting to undergo the process of replanting (Danylo et al., 2021; Numata et al., 2022) although replanting in smallholder plantations is often delayed as farmers face more financial constraints (Zhao et al., 2023). Hence, we expect to see a more complex influence of OP on LAI trend depending on the management practices of different types of plantations.



In comparison with CRO, our findings indicate that the impact of OP expansion on vegetation greenness decline is relatively minor (Fig. 5). This can be attributed to the generally higher biomass of OP compared to typical crops like rice and maize in Indonesia and Malaysia. It is also important to note that our estimation of CRO or OP expansion was based on the assumption that the increased areas of CRO or OP since 2001 came from EBF. While CRO and OP expansion indeed mostly resulted from

deforestation in Indonesia and Malaysia (Numata et al., 2022; Wagner et al., 2022), the expansions of CRO and OP can also come from other land uses (i.e., grasslands or pastures). Transitions from these land uses to CRO and OP might result in a smaller negative impact on vegetation greenness, we suspect, considering grasslands and shrubs have smaller LAI than EBF.

### 4.3 Other LUCC impacts on greenness in Southeast Asia

While our study examined the two most prominent processes of LUCC in Southeast Asia (EBF to OP and EBF to CRO), there

are other types of land use changes we analyzed together under the category of "Other" land use types. The changes in these land use types are also relevant to deforestation, and they include other plantations such as rubber plantations (Wang et al., 2023), agroforests such as cocoa and coffee (Pantera et al., 2021), and grasslands or pastures (Austin et al., 2019). In total, "Other" accounted for 4.70% of the study area in 2016, much less than the three main types we studied (51.06% for EBF, 25.01% for CRO and 12.09% for OP in 2016), and experienced minor changes since 2001.

Meanwhile, we found the overall impact of these other LUCC on greenness was likely small (Fig. 5f). The small impact is contingent on their smaller extent compared to EBF, CRO and OP (see above). It may also result from the offset of positive and negative impacts from individual Other LUCC processes on greenness. For example, rubber plantations exhibit a higher LAI than natural forests (Wang et al., 2022b), agroforestry and other plantations generally have a lower LAI than nature forests, therefore leading to different trends in LAI after deforestation.

### 305 4.4 Impact of climate change and $CO_2$ concentrations on regional greenness

$CO_2$ fertilization effects appear to be the primary drivers of greening trends observed in global studies (Chen et al., 2022; Ewert, 2004; Zhu et al., 2016). Our research also confirmed the substantial contribution of rising CO2 levels to the greening of vegetation in Southeast Asia. The impact of temperature on greenness in Southeast Asia was negative, in contrast to the positive effects noted in cold climate zones, such as the Qinghai-Tibet Plateau (Zhong et al., 2019) and Arctic areas (Forbes

et al., 2010). We suspect that the negative effect of temperature implies that temperature may have exceeded the optimal point for plant growth in parts of Southeast Asia. This aligns with several studies suggesting that ecosystem functions in the tropics are approaching a temperature tipping point (Doughty et al., 2023; Meir et al., 2015; Wu et al., 2019). Additionally, temperature rise might exacerbate the incidence of pests and diseases in tropical forests, negatively impacting plant health and productivity (Ghini et al., 2011). These factors might contribute jointly to the observed decline in greenness with increasing temperatures

in Southeast Asia.

**5. Conclusion**

Our study closely examined the impacts of LUCC on vegetation greenness in part of Southeast Asia. We found that there was no significant trend in vegetation greenness in the study area, which is attributed to the net effect of negative impacts of LUCC on LAI and positive influence of elevated $CO_2$ on LAI. Among various LUCC processes, we found that cropland expansion

was the primary reason for LAI decrease, while oil palm expansion had a small impact on LAI trends. These results shed light on the interplay between greenness and land use changes and provide valuable insight into our future studies on terrestrial carbon, water and energy budgets in the land use change intensive Southeast Asia.

**Acknowledgment**

XL and RZ are supported by a Tier 2 research grant from Singapore Ministry of Education (A-8001551-00-00) and a Singapore

Energy Centre Core project (A-8000179-00-00). We also thank Tin Widyani Satriawan, a PhD student in our group, for her valuable suggestions on refining the manuscript's language.

**Data Availability**

The GLOBMAP LAI dataset is obtained from https://zenodo.org/records/4700264. Climatic variables including temperature, precipitation, wind speed, shortwave downward radiation, and humidity were obtained from the ERA5 model

(https://cds.climate.copernicus.eu/cdsapp#!/dataset/reanalysis-era5-single-levels?tab=form). The distribution of forest age and plantation age maps was obtained from the (https://sedac.ciesin.columbia.edu/data/collection/gpw-v4). $CO_2$ concentrations are obtained from (https://gml.noaa.gov/dv/data/).

**Code Availability**

The code used for this research will be made publicly available upon acceptance of the manuscript.

**Author contribution**

XL and RZ conceptualized the study, visualization, and writing of the original draft preparation. RZ and YY contributed to the data curation, methodology, and software development. RZ, XL, and YY performed the formal analysis. XL acquired the funds. LS, CC and JL were behind the project administration and the supervision of the research planning. RZ and YY contributed to the software development. All authors contributed to the writing review, editing, and validation.



**Competing interest**

The authors declare no competing interests.

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
