# Peer review of "Cropland expansion drives vegetation greenness decline in Southeast Asia"

_EGUsphere, 2024_

## Referee Comment (RC1)

The manuscript effectively quantifies the contributions of various factors to land surface greening and LAI changes. The aim of the manuscript is clearly articulated, and the background for the problem in the study area is well presented. However, I have some concerns regarding the methods and results that need clarification for further consideration. Below are my major and minor comments:

1. About the LUCC detection methods

The main conclusion of the manuscript is that cropland expansion acts as the primary factor offsetting the greening trend resulting from climate change and $CO_2$ elevation. While this finding is good, I have big concerns regarding the robustness of the input data used in this manuscript. The authors take the percentiles of crop expansion in 500m grid cells equal to the available 0.25-degree cells introduced many uncertainties. Representing higher resolution grid cells with coarser ones is unconventional. In this way, it is not able to identify differences among individual 500m pixels, further complicating the analysis.

A common technique for addressing such challenges in remote sensing is data fusion, although it requires many additional efforts. One potential solution to enhance the robustness of the manuscript's findings could involve conducting analyses using a matched resolution of 0.25 degrees rather than 500m. This adjustment would mitigate uncertainties associated with the coarse-to-fine representation of data.

Moreover, the methodology described in section 2.3 is long with text. it would be beneficial to include a conceptual figure illustrating a comparison of pixels. This would improve the clarity of the methodology section.

Additionally, when upscaling global forest change maps from 30m to 500m resolution, the issue of non-integer pixel numbers within a 500m grid cell arises. How did you treat the boundary cells? Clarification and a clear explanation are needed regarding how the methodology addresses boundary cells for the analysis.

2. About the Machining learning Approaches

Machine learning methodologies, however, have often remained as "black boxes" to ecologists due to their intricate algorithmic nature and limited interpretability regarding their predictive power (Simon, Glaum and Valdovinos 2023). In the manuscript, the authors establish five scenarios for predicting Leaf Area Index (LAI) by maintaining certain variables unchanged. However, it remains unclear how the machine learning method treats the unchanged variables within each scenario. Furthermore, the manuscript lacks description on the disparities in predicted LAI across scenarios, particularly concerning the inclusion of varying changed variables. It is important for the authors to select specific pixels to show the gradual changes in prediction results and accuracy, from the first, second, and to the last scenario.

In Figure 3, the specific scenario being shown here is not mentioned. Clarification regarding which scenario is represented in the figure is necessary. Additionally, inclusion of a time-series curve, focusing on a specific pixel, would provide valuable insight into the predictions generated by the machine learning algorithms.

Moreover, the manuscript fails to show details about splitting the data into training and testing sets, and how different splitting ratios may impact the conclusions drawn. A comprehensive explanation of the data splitting process and its potential implications on the study's findings is essential.

3. About the LAI change trend

I have concerns regarding the GLOBMAP LAI dataset, which is also the only LAI product used in the manuscript. There is substantial variability among global LAI products. However, it is notable that even most LAI datasets depict an increasing trend in LAI changes, the GLOBMAP dataset stands out as an exception, which characterized by notably lower values (Jiang, Ryu et al. 2017), see the figure below.

The slight change trend observed in the LAI within the study area could potentially be attributed to the specific LAI products utilized by the authors. To enhance the confidence in the conclusions, the inclusion of additional LAI products is essential to provide a more comprehensive assessment of vegetation dynamics.

4. About the assessing trend contribution from variables

In section 3.3, the authors outlined their approach to calculating the contributions of various variables to LAI trends. However, there appears to be confusion regarding the methodology's assessment. As stated, did the authors evaluate the contribution of elevated $CO_2$ to greening by comparing the LAI trend from Scenario 1 to that from Observation? If so, authors should present scatterplots and regression lines for each scenario, and their statistical significance to allowing readers to know the differences among them.

In addition, it seems the authors calculated the average value of the trend for all the pixels as the final conclusions. How did you examine the significant of the machine learning results for each pixel before calculating the mean value? Furthermore, considering pixels with no significance, how might this bias the conclusion? It's crucial for the authors to address these concerns to ensure the reliability of their findings.

[Figure]

**Figure 1**

Global mean LAI (solid curves) and linear trends during 1982–2011 (dotted lines), 1982–1999 (long dashed lines), and 2003–2011 (short dashed lines) for different LAI products. Trend values are listed in Table 2. Lifespans of different satellite platforms are also illustrated

Jiang, C., et al. (2017). "Inconsistencies of interannual variability and trends in long-term satellite leaf area index products." Global Change Biology **23**(10): 4133-4146.

Simon, S. M., et al. (2023). "Interpreting random forest analysis of ecological models to move from prediction to explanation." Scientific Reports **13**(1): 3881.

---

## Referee Comment (RC2)

This manuscript provides a detailed analysis of how land use changes in the Southeast Asia region affect vegetation greenness. Utilizing multi-source land cover datasets, it reveals how the transformation of land use since the 21st century has impacted vegetation greenness, based on machine learning algorithms and the SHAP interpreter. The topic of this manuscript is interesting, explaining why China and India, despite both being countries with rapidly developing agriculture, make significant contributions to greening trends, while the greening trend in Southeast Asia remains stagnant. However, certain aspects may need addressing before publication.

**Major comments**

1. The literature review concerning the driving mechanisms behind vegetation greenness changes in Southeast Asia appears to be incomplete and insufficiently detailed. It is essential to provide a more comprehensive overview of existing research to adequately situate the study within the current body of knowledge.

2. The methodology section requires significant revision due to several critical issues:

   - The use of citations is improper, with several missing references that need to be included to support the study's claims and methodology.

   - Details regarding the specific version of the dataset used and the preprocessing steps undertaken are absent, which is crucial for the reproducibility and integrity of the research.

   - The explanation of how multiple land cover datasets were harmonized lacks clarity, making it difficult to understand the approach taken.

   - Descriptions of scenario simulations are unclear. When introducing scenario simulation schemes, it is imperative to explicitly detail the calculation methods for assessing the impact of each factor, which would greatly enhance the manuscript's credibility and reliability.

**Minor comments**

1. The discussion mentions, "It is also important to note that our estimation of CRO or OP expansion was based on the assumption that the increased areas of CRO or OP since 2001 came from EBF." Such a crucial assumption should be stated in the methodology section.

2. On the basis of Figure 5, it would be beneficial to add the spatial distribution of dominant factors for each pixel. This enhancement would more clearly reveal whether the LAI trend for each pixel is positive or negative and which factors primarily drive these changes.

3. It would be preferable to represent Figure S2 as a scatter density plot (like Figure 4c,d) to facilitate the observation of changes in SHAP values with features, and to prevent potential misinterpretation arising from the clustering of scatter points.

4. There is an error in Equation (2) that needs to be corrected.

5. Figure 4c,d depicts the coupling effects of f_EBF with f_OP and f_CRO rather than the interaction effects mentioned in the caption, making it seem indistinguishable from Figure S2a. It is recommended to add SHAP dependence plots illustrating the interaction effects for a more in-depth analysis.

6. Previous studies have highlighted discrepancies between the cropland area changes provided by LUH2 and actual conditions in China and the United States. It is worth investigating whether a similar discrepancy exists in Southeast Asia. Meanwhile, the spatial resolution of the LUH2 dataset is too coarse for the purposes of this study.

---

## Author Comment (AC1)

Reviewer #1

**R1C1:** The manuscript effectively quantifies the contributions of various factors to land surface greening and LAI changes. The aim of the manuscript is clearly articulated, and the background for the problem in the study area is well presented. However, I have some concerns regarding the methods and results that need clarification for further consideration. Below are my major and minor comments:

**Response:** We appreciate the positive and constructive comments from the reviewer. In this round of revision, we have endeavored to take their comments on board to improve the manuscript. The primary changes include the addition of a workflow diagram and details about the datasets we used. We also included a selected pixel to better illustrate our scenario simulations. Please kindly refer to our point-to-point response below.

**R1C2:** 1. About the LUCC detection methods

The main conclusion of the manuscript is that cropland expansion acts as the primary factor offsetting the greening trend resulting from climate change and CO2 elevation. While this finding is good, I have big concerns regarding the robustness of the input data used in this manuscript. The authors take the percentiles of crop expansion in 500m grid cells equal to the available 0.25-degree cells introduced many uncertainties. Representing higher resolution grid cells with coarser ones is unconventional. In this way, it is not able to identify differences among individual 500m pixels, further complicating the analysis.

A common technique for addressing such challenges in remote sensing is data fusion, although it requires many additional efforts. One potential solution to enhance the robustness of the manuscript's findings could involve conducting analyses using a matched resolution of 0.25 degrees rather than 500m. This adjustment would mitigate uncertainties associated with the coarse-to-fine representation of data.

**Response:** We appreciate the detailed guidance from the reviewer. Following their suggestion, we reconducted our analyses using a matched resolution of 0.25 degrees (Fig. S8 below). The new results indicated that the spatial pattern of land use and the changes over time were similar to the original results we obtained at a finer resolution (Fig. 2, Fig. 6), suggesting our result were robust to the choice of spatial resolution.

[Figure]

**Figure S8: Land use composition and its changes from 2001 to 2016 in the study area, analyzed at a 0.25-degree resolution.**

[Figure]

**Figure 2: Land use composition and its changes from 2001 to 2016 in the study area, analyzed at a 500-m resolution.**

[Figure]

**Figure S6: The spatial distribution of the pixel-wise impacts of each process on the greening trends, analyzed at a 0.25-degree resolution.**

[Figure]

**Figure 6: The spatial distribution of the pixel-wise impacts of each process on the greening trends, analyzed at 500-m resolution.**

**R1C3:** Moreover, the methodology described in section 2.3 is long with text. it would be beneficial to include a conceptual figure illustrating a comparison of pixels. This would improve the clarity of the methodology section.

**Response:** We have included a conceptual figure (Fig. S1) to clarify how we compared and harmonized multiple land cover datasets, according to the suggestion from the reviewer.

[Figure]

**Figure. S1: A conceptual figure illustrating the processes of harmonizing land cover datasets with in a grid with a spatial resolution of 500 meter in this study.** Step (1): upscale fine-resolution global forest change maps (GFC) to 500 meter to determine forest (A%) and non-forest (B%). Step (2) and step (3): calculate percentages of oil palm, evergreen broadleaf forest (EBF), and other forest types (OT1) within forest areas. Oil palm percentages are derived by upscaling a 100-meter resolution oil palm product to 500 meters. EBF and OT1 percentages are sourced from the MODIS dataset. Step (4): Determine cropland and other land uses (OT2) percentages using the LUH2 dataset, assuming LUH2 data at a 0.25° grid applies to 500-meter grid cells within each 0.25° grid cell. Note, the conceptual figure illustrates only the percentage of each land use, not their specific locations.

In addition, we have added a flowchart (Fig. 1) to this section to provide a step-by-step illustration on our method. We added the numbers of each step and detailed description from Line xx to Line xx for further clarification: "As shown in Fig. 1, the workflow for harmonizing multiple land cover datasets involved the following steps:

(1) We first determined the annual percentage of forested (A%) and non-forested areas (B%) within each 500m grid cell by aggregating the mean of the annual 30 m resolution Global Forest Change v1.11 (GFC) maps (Hansen et al., 2013).

(2) Within the forested fraction of each grid cell, we estimated the proportion of oil palm (OP) plantations (A1%) using an openly available dataset that covers OP distribution from 2001 to 2016 across Malaysia and Indonesia (Xu et al., 2020). To estimate the proportion of OP, we calculated the frequency of oil palm pixels in each 500 m × 500 m window.

(3) After accounting for the area of OP, the remaining forested area in each grid was further categorized into the evergreen broadleaf forest (EBF) (A2%) and other forest types (i.e., deciduous broadleaf forest, coniferous forest, mixed forest, etc.), based on the ratio of EBF to the total forested area provided by MODIS Land Cover Type Product (MCD12Q1 v6.1) (Sulla-Menashe and Friedl, 2018).

(4) Within the non-forested fraction of each grid cell, we used the latest version of the Land-use harmonization datasets (LUH2) dataset (Hurtt et al., 2020) to estimate the percentage of cropland (CRO) (B1%) and other non-forest land uses (i.e., pasture, grass, etc.). In this analysis, we assumed that the fraction of each land use type in

the LUH2 dataset on a 0.25° grid is applicable to the 500 m grid cells within each 0.25° grid cell.

At the end, we obtained detailed information for EBF, OP, CRO, and "Other" land-use types (including other forests and non-forest vegetated areas), at the 500 m spatial resolution. We grouped other forests and other non-forest vegetated areas together, as they represented a minor proportion (less than 5%) of the land surface (Table S2) and exhibited minimal changes during the study period."

[Figure]

**Figure 1: Workflow of the study.** Steps (1) to (4) outline the processes for harmonizing multiple land cover datasets. Steps (5) to (6) show the establishment and interpretation of the LAI prediction machine learning model and the process of scenario simulations.

**R1C4:** Additionally, when upscaling global forest change maps from 30m to 500m resolution, the issue of non-integer pixel numbers within a 500m grid cell arises. How did you treat the boundary cells? Clarification and a clear explanation are needed regarding how the methodology addresses boundary cells for the analysis.

**Response:** We agree with the reviewer that it is very likely to have some non-integer pixel numbers over boundaries when upscaling high resolution map to low resolution – in this case from global forest change maps at 30m to 500m resolution. We addressed this issue by using the 'reduceResolution()' function, the default aggregation method in Google Earth Engine (GEE). According to GEE, the weights assigned to pixels during the aggregation process are determined by the extent of overlap between the smaller pixels being aggregated and the larger pixels defined by

the output projection. This is illustrated in Figure R1. In the diagram, the output pixel has area a (i.e., 500m × 500m in our study), the weight of the input pixel with intersection area b is computed as b/a, and area c is computed as c/a. To compute forested area per pixel, use the fraction of a pixel covered, then multiply by area.

To make it clear for readers, we improved our statements about the upscaling process from Line 100 to Line 105: "We first determined the annual percentages of forested (A%) and non-forested areas (B%) within each 500m grid cell by aggregating annual 30m resolution Global Forest Change v1.11 (GFC) maps (Hansen et al. 2013), using the 'reduceResolution()' function in Google Earth Engine (https://developers.google.com/earth-engine/guides/resample)."

[Figure]

**Figure R1: Input pixels (black) and output pixel (blue) for reduceResolution() in google earth engine.** Source: https://developers.google.com/earth-engine/guides/resample

**R1C5:** 2. About the Machining learning Approaches

Machine learning methodologies, however, have often remained as "black boxes" to ecologists due to their intricate algorithmic nature and limited interpretability regarding their predictive power (Simon, Glaum and Valdovinos 2023). In the manuscript, the authors establish five scenarios for predicting Leaf Area Index (LAI) by maintaining certain variables unchanged. However, it remains unclear how the machine learning method treats the unchanged variables within each scenario. Furthermore, the manuscript lacks description on the disparities in predicted LAI across scenarios, particularly concerning the inclusion of varying changed variables. It is important for the authors to select specific pixels to show the gradual changes in prediction results and accuracy, from the first, second, and to the last scenario.

**Response:** We appreciate the reviewer's suggestions. Accordingly, we randomly selected a pixel (102.15°E, 0.95°S) to illustrate the changes in predicted LAI across scenarios (Fig. S2). Specifically, in scenario simulations, we adjusted the input variables according to specific assumptions to progressively quantify the impact of different processes on the greening trend.

For S1 (indicated by the black line), we assumed that only $CO_2$ varied from 2001 to 2016, while climate and land use variables (i.e., CLI, f_EBF, f_CRO, f_OP and f_Other) remained constant at their values in 2001. This scenario simulated the greening trend (i.e., $\beta1 = 0.07$) solely attributed to elevated $CO_2$ concentration.

For S2 (indicated by the red line), we assume $CO_2$ concentration and climate change (CLI) over time, with land uses remaining unchanged since 2001. The difference between the trends in S2 and S1 is attributed to the impact of CLI on the greening trend (i.e., $\beta2 - \beta1$).

S3 to S5 sequentially considered different land use processes. S3 (indicated by the blue line) involved changes from EBF to CRO and time-varying $CO_2$ and climate, while keeping OP and other land use types constant post-2001; The difference between S3 and S2 highlights the impact of CRO expansion on the greening trend (i.e., $\beta3 - \beta2$), showing a significant decrease (around -0.24).

S4 (indicated by the green line) included conversions from EBF to both CRO and OP with time-varying $CO_2$, climate, while other land uses unchanged since 2001; The difference between S4 and S3 illustrates the minimal impact of OP expansion on the greening trend (i.e., $\beta4 - \beta3$).

S5 encompassed all LUCC changes, with all variables including $CO_2$, climate, and all types of LUCC varying over time. The different trends between S5 and S4 indicate impact of other LUCC on greening trend (i.e., $\beta5 - \beta4$).

[Figure]

**Figure S2: A selected pixel to show the gradual changes in prediction results for each scenario.**

**R1C6:** In Figure 3, the specific scenario being shown here is not mentioned. Clarification regarding which scenario is represented in the figure is necessary. Additionally, inclusion of a time-series curve, focusing on a specific pixel, would

provide valuable insight into the predictions generated by the machine learning algorithms.

**Response:** Thank you for your comment. In Figure 3, we presented the calibration and validation results of the machine learning model, not a hypothetical scenario (we could not do validation for hypothetical scenarios as hypothetical scenarios do not have ground truth).

Perhaps to further clarify, our study consisted of two main steps: (1) Establishing the relationship between LAI and environmental factors using machine learning algorithms. This is what Figure 3 illustrates. (2) Scenario simulations by applying the established machine learning model to predict the impacts of various processes ($CO_2$ elevation, climate change, CRO expansion, OP expansion, and other LUCC) on vegetation greenness. Please refer to the workflow chart in our response to R1C3, where the model establishment and scenario predictions are detailed in Steps 5 and 6, respectively. Regarding the time-series curve illustrating predictions generated by our machine-learning method, please see our response to R1C5.

**R1C7:** Moreover, the manuscript fails to show details about splitting the data into training and testing sets, and how different splitting ratios may impact the conclusions drawn. A comprehensive explanation of the data splitting process and its potential implications on the study's findings is essential.

**Response:** Following previous studies (Wang et al., 2022; Abel et al., 2023), we randomly split the data into training and testing sets with a ratio of 80%:20% in our study. We have added the details in our manuscript from Line 165 to Line 166. To assess the impact of different splitting ratios, we also tested 70%:30% and 60%:40% ratios. We found that these different splitting ratios had minimal impact on the model performance and interpretations.

[Figure]

**Figure S3: The impact of different training and testing dataset splitting ratios on model performance and interpretations.** Panels (a) and (d) depict results using an 80%:20% ratio for training and testing, respectively. Panels (b) and (e) correspond to a 70%:30% ratio, while panels (c) and (f) reflect a 60%:40% ratio.

Wang H, Yan S, Ciais P, et al. Exploring complex water stress–gross primary production relationships: Impact of climatic drivers, main effects, and interactive effects[J]. Global Change Biology, 2022, 28(13): 4110-4123.

Abel C, Abdi A M, Tagesson T, et al. Contrasting ecosystem vegetation response in global drylands under drying and wetting conditions[J]. Global change biology, 2023, 29(14): 3954-3969.

**R1C8:** 3. About the LAI change trend

I have concerns regarding the GLOBMAP LAI dataset, which is also the only LAI product used in the manuscript. There is substantial variability among global LAI products. However, it is notable that even most LAI datasets depict an increasing trend in LAI changes, the GLOBMAP dataset stands out as an exception, which characterized by notably lower values (Jiang, Ryu et al. 2017), see the figure below.

[Figure]

**Figure 1**    Open in figure viewer | ⬇PowerPoint

Global mean LAI (solid curves) and linear trends during 1982–2011 (dotted lines), 1982–1999 (long dashed lines), and 2003–2011 (short dashed lines) for different LAI products. Trend values are listed in Table **2**. Lifespans of different satellite platforms are also illustrated

The slight change trend observed in the LAI within the study area could potentially be attributed to the specific LAI products utilized by the authors. To enhance the confidence in the conclusions, the inclusion of additional LAI products is essential to provide a more comprehensive assessment of vegetation dynamics.

**Response:** Thanks for pointing this out, but we would like to clarify a neglected issue in the community. The GLOBEMAP LAI product we used is the most recent version (version 3) and has been tested to be robust for global greening trend studies (Piao et al., 2020; Winkler et al., 2021). The GlOBEMAP LAI used in Jiang's study was a very preliminary version and could not indicate the true performance of the dataset. Made fully public since 2021 (https://doi.org/10.5281/zenodo. 4700264), the GLOBEMAP LAI involved several improvements (Liu et al., 2021): it employed MODIS C6 land surface reflectance products (MOD09A1) for generating MODIS LAI, accounted for pixel-level clumping effects, and utilized a new cloud detection algorithm. This updated version of LAI product showed an increasing trend globally, consistent with other datasets (Fig. R2).

We use GLOBEMAP v3 for two reasons: (1) it is a primary LAI dataset for global and regional greenness studies (Piao et al., 2020; Winkler et al., 2021; Satriawan et al., 2024), showing high consistency with other LAI datasets; (2) it is generated with an advanced algorithm to consider canopy clumping, making it particularly suitable for dense canopies in the tropics (Fang et al., 2019).

In addition to enhancing reliability, we further analysed the greening trend using MODIS LAI datasets (Fig. S4). Our findings indicated that both GLOBEMAP and MODIS LAI datasets demonstrated a moderate increasing trend across the entire region (Fig. S4a). For pixel-by-pixel validation, over 70% of the regions exhibited a

consistent trend (Fig. S4b), with very similar spatial pattern of the trend. We have included the figure and corresponding statements in the Supplementary file from Line xx to Line xx. We noticed that the annual change in LAI in the region show much larger interannual variation, which is very untypical for tropical ecosystems. Therefore in the main analysis, we continue to use GLOBEMAP LAI but added the results based on MODIS in SI.

[Figure]

**Figure R2: Changes in satellite-derived global vegetation indices from four products: GIMMS, GLASS, GLOBMAP and MODIS.** Sources: (a) is from Piao et al (2020), and the panel (b) is from Winkler et al (2021).

[Figure]

**Figure S4: Comparison of LAI Trends Between MODIS and GLOBEMAP LAI Datasets.** (a) illustrates the relative changes in annual mean LAI across the entire region from 2001 to 2016. (b) provides a spatial comparison of the datasets, where '++' denotes an increase observed in both datasets, '−−' indicates a decrease in both, '+−' signifies an increasing trend in GLOBEMAP but a decrease in MODIS, and '−+' represents the opposite scenario.

References:

Liu, R., Liu, Y., & Chen, J. (2021). GLOBMAP global leaf area index since 1981 (3.0) [Dataset]. Zenodo. https://doi.org/10.5281/zenodo. 4700264

Piao, S., Wang, X., Park, T., Chen, C., Lian, X., &He, Y., et al. (2020). Characteristics, drivers and feedbacks of global greening. Nature reviews. Earth & environment, 1(1), 14-27. http://doi.org/10.1038/s43017-019-0001-x

Winkler A J, Myneni R B, Hannart A, et al. Slowdown of the greening trend in natural vegetation with further rise in atmospheric $CO_2$. Biogeosciences, 2021, 18(17): 4985-5010.

Satriawan T W, Luo X, Tian J, et al. Strong green‐up of tropical Asia during the 2015/16 El Niño[J]. Geophysical Research Letters, 2024, 51(8): e2023GL106955.

Fang, H., Baret, F., Plummer, S., &Schaepman Strub, G. (2019). An Overview of Global Leaf Area Index (LAI): Methods, Products, Validation, and Applications. Reviews of Geophysics, 57(3), 739-799. http://doi.org/10.1029/2018RG000608

**R1C9**: 4. About the assessing trend contribution from variables

In section 3.3, the authors outlined their approach to calculating the contributions of various variables to LAI trends. However, there appears to be confusion regarding the methodology's assessment. As stated, did the authors evaluate the contribution of elevated CO2 to greening by comparing the LAI trend from Scenario 1 to that from Observation? If so, authors should present scatterplots and regression lines for each scenario, and their statistical significance to allowing readers to know the differences among them.

**Response:** We apologize for the confusion regarding the statements of the methodology part. As addressed in our response to R1C6, the contribution of elevated $CO_2$ to greening was estimated using scenario simulations (i.e., step 6 in our workflow). This step involved hypothesis testing for attribution analysis and thus could not be validated using observations. Specifically, we assumed that only $CO_2$ varied over time, with no climate and land-use change changes since 2001. Consequently, the trend in LAI in this scenario was thus solely attributed to $CO_2$ variations.

**R1C10:** In addition, it seems the authors calculated the average value of the trend for all the pixels as the final conclusions. How did you examine the significant of the machine learning results for each pixel before calculating the mean value? Furthermore, considering pixels with no significance, how might this bias the conclusion? It's crucial for the authors to address these concerns to ensure the reliability of their findings.

**Response:** In our final conclusions regarding the impact of each process on the regional LAI trend (i.e., Fig. 6f), we first calculated the annual mean LAI for the entire region for each hypothesis scenario. We then analyzed the trends. The differences in trends between scenarios represented the contribution of each process to the overall regional LAI trend. At this stage, we did not exclude the non-significant pixels, because we treat the study area as a whole.

For the pixel-level analysis of the impact of each process on LAI trend (i.e., Fig. 6a –
6e), we provided the significance ($p < 0.01$) of trends under different scenarios (Fig.
S5). But when comparing trends between scenario simulations, we included non-
significant pixels. This approach was adopted because (1) trends of LAI in one pixel
may be significant under one scenario, but not significant in another, therefore it
would not be apple to apple comparisons if we only use different number of pixels
from different scenarios for comparison; (2) excluding non-significant pixels could
potentially overestimate the impact of specific processes. For example, CRO
expansion might increase the trend in some pixels without reaching statistical
significance. Ignoring such pixels could lead to overestimating the negative impact of
CRO expansion on greening trends.

[Figure]

**Figure 6: The spatial distribution of the pixel-wise impacts of each process on the greening
trends.**

[Figure]

**Figure S5: The spatial distribution of the pixel-wise impacts of each scenario on the greening
trends.**

---

## Author Comment (AC2)

Reviewer #2

**R2C1:** This manuscript provides a detailed analysis of how land use changes in the Southeast Asia region affect vegetation greenness. Utilizing multi-source land cover datasets, it reveals how the transformation of land use since the 21st century has impacted vegetation greenness, based on machine learning algorithms and the SHAP interpreter. The topic of this manuscript is interesting, explaining why China and India, despite both being countries with rapidly developing agriculture, make significant contributions to greening trends, while the greening trend in Southeast Asia remains stagnant. However, certain aspects may need addressing before publication.

**Response:** We appreciate the accurate summary and positive comments from the reviewer, and thank them for recognizing the importance of our work on studying regional greening trends.

Major comments

**R2C2:** The literature review concerning the driving mechanisms behind vegetation greenness changes in Southeast Asia appears to be incomplete and insufficiently detailed. It is essential to provide a more comprehensive overview of existing research to adequately situate the study within the current body of knowledge.

**Response:** According to the suggestion, we further enhanced the literature review on the drivers of vegetation greenness in Southeast Asia. It is unfortunate that few studies specifically focused on this region (e.g., Satriawan et al., 2024), and we gained most of our knowledge on Southeast Asia from global scale studies (e.g., Zhu et al., 2016; Piao et al., 2019; Chen et al., 2019; Chen et al., 2022). Specifically, these global studies reveal that $CO_2$ fertilization is a primary driver of the greening trend globally, including in Southeast Asia (Zhu et al., 2016, Chen et al., 2022). Climate change, especially temperature rise, could reduce vegetation growth in the tropics (Piao et al., 2019) or drive green-up in maritime Southeast Asia during El Niño (Satriawan et al., 2024). However, land-use change, especially deforestation, is the predominant factor driving the greenness decline in tropical countries like Indonesia (Piao et al., 2019; Chen et al., 2019).

We have included the corresponding references in our manuscript and the context is added in Line 35 to Line 45, "Southeast Asia harbours diverse biodiversity and ecosystems. Yet, the trends and drivers of regional greenness remain largely underexplored. Previous studies reveal that $CO_2$ fertilization is a primary driver of the greening trend in Southeast Asia within a global context (Zhu et al., 2016, Chen et al., 2022). The impact of climate change on vegetation growth, however, remains uncertain (Piao et al., 2019, Satriawan et al., 2024), although some studies have reported that tropical temperature approaching critical thresholds may lead to leaf browning (Doughty et al., 2023). Land-use change, particularly deforestation, has

been found to be a predominant factor causing the decline of greenness in some tropical regions (Piao et al., 2019; Chen et al., 2019). However, these studies primarily focused on a global scale while the regional mechanisms (i.e., complexity in land use change) for greenness change were not fully examined"

Satriawan T W, Luo X, Tian J, et al (2024). Strong green‑up of tropical Asia during the 2015/16 El Niño. Geophysical Research Letters, 51(8): e2023GL106955

Zhu, Z., Piao, S., Myneni, R. B., Huang, M., Zeng, Z., &Canadell, J. G., et al. (2016). Greening of the Earth and its drivers. Nature Climate Change, 6(8), 791-795. http://doi.org/10.1038/nclimate004

Piao, S., Wang, X., Park, T., Chen, C., Lian, X., &He, Y., et al. (2019). Characteristics, drivers and feedbacks of global greening. Nature reviews. Earth & environment, 1(1), 14-27. http://doi.org/10.1038/s43017-019-0001-x

Chen, C., Park, T., Wang, X., Piao, S., Xu, B., &Chaturvedi, R. K., et al. (2019). China and India lead in greening of the world through land-use management. Nature Sustainability, 2(2), 122-129. http://doi.org/10.1038/s41893-019-0220-7

Chen, C., Riley, W. J., Prentice, I. C., &Keenan, T. F. (2022). $CO_2$ fertilization of terrestrial photosynthesis inferred from site to global scales. Proceedings of the National Academy of Sciences, 119(10). http://doi.org/10.1073/pnas.2115627119

Chen, J. M., &Black, T. A. (1992). Defining leaf area index for non‑flat leaves. Plant, cell and environment, 15(4), 421-429. http://doi.org/10.1111/j.1365-3040.1992.tb00992.x

**R2C3:** The methodology section requires significant revision due to several critical issues: · The use of citations is improper, with several missing references that need to be included to support the study's claims and methodology.

**Response:** We double-checked the citations in the methodology section and added the necessary references. Specifically, we added references (i.e., Euler et al., 2016; Chen et al., 2024) to support our statements about land use change in Southeast Asia, and references (i.e., Fable, 2020; Sulla-Menashe and Friedl, 2018; Hurtt et al., 2020; Lundberg et al., 2018; Sitch et al., 2015) about the methodology regarding harmonization of different land use datasets, and the XGBoost-SHAP framework and scenario simulations (i.e., Lundberg et al., 2018; Sitch et al., 2015).

Euler M, Schwarze S, Siregar H, et al. Oil palm expansion among smallholder farmers in Sumatra, Indonesia[J]. Journal of Agricultural Economics, 2016, 67(3): 658-676.

Chen S, Woodcock C, Dong L, et al. Review of drivers of forest degradation and deforestation in Southeast Asia[J]. Remote Sensing Applications: Society and Environment, 2023: 101129.

Fable (2020). Pathways to Sustainable Land-Use and Food Systems. 2020 Report of the FABLE Consortium. International Institute for Applied Systems Analysis (IIASA) and Sustainable Development Solutions Network (SDSN), Laxenburg and Paris. 10.22022/ESM/12-2020.16896. Indonesia chapter.

Sulla-Menashe D, Friedl M A. User guide to collection 6 MODIS land cover (MCD12Q1 and MCD12C1) product[J]. Usgs: Reston, Va, Usa, 2018, 1: 18.

Hurtt G C, Chini L, Sahajpal R, et al. Harmonization of global land use change and management for the period 850–2100 (LUH2) for CMIP6[J]. Geoscientific Model Development, 2020, 13(11): 5425-5464.

Lundberg S M, Erion G G, Lee S I. Consistent individualized feature attribution for tree ensembles[J]. arXiv preprint arXiv:1802.03888, 2018.

Sitch S, Friedlingstein P, Gruber N, et al. Recent trends and drivers of regional sources and sinks of carbon dioxide[J]. Biogeosciences, 2015, 12(3): 653-679.

**R2C4:** · Details regarding the specific version of the dataset used and the preprocessing steps undertaken are absent, which is crucial for the reproducibility and integrity of the research.

**Response:** We apologize for the missing information about the version of the dataset we used. In the updated manuscript, we have included details about GLOBMAP LAI (v3), Global forest change maps (v1.11), and MODIS land cover product (v6.1). We have also made sure that version numbers for other land use datasets and climate datasets used in our study were provided.

We hope to clarify that the datasets we collected are all established products. The main processing we carried out is to harmonize them into a common grids. To better describe the process, we have included a figure and statements in SI (see below).

[Figure]

**Figure. S1: A conceptual figure illustrating the processes of harmonizing land cover datasets with in a grid with a spatial resolution of 500 meter in this study.** Step (1): upscale fine-resolution global forest change maps (GFC) to 500 meter to determine forest (A%) and non-forest (B%). Step (2) and step (3): calculate percentages of oil palm, evergreen broadleaf forest (EBF), and other forest types (OT1) within forest areas. Oil palm percentages are derived by upscaling a 100-meter resolution oil palm product to 500 meters. EBF and OT1 percentages are sourced from the MODIS dataset. Step (4): Determine cropland and other land uses (OT2) percentages using the LUH2 dataset, assuming LUH2 data at a 0.25° grid applies to 500-meter grid cells within each 0.25° grid cell. Note, the conceptual figure illustrates only the percentage of each land use, not their specific locations.

**R2C5:** · The explanation of how multiple land cover datasets were harmonized lacks clarity, making it difficult to understand the approach taken.

**Response:** To enhance clarity, we moved the flowchart that illustrates how multiple land cover datasets were harmonized from the supplementary file to the Methods section. Additionally, we added order numbers and corresponding statements from Line 100 to Line 115 in this section for further clarification: "As shown in Fig. 1, the workflow for harmonizing multiple land cover datasets involved the following steps:

(1) We first determined the annual percentage of forested (A%) and non-forested areas (B%) within each 500m grid cell by aggregating the mean of the annual 30 m resolution Global Forest Change v1.11 (GFC) maps (Hansen et al., 2013), using the 'reduceResolution()' function in Google Earth Engine (https://developers.google.com/earth-engine/guides/resample).

(2) Within the forested fraction of each grid cell, we estimated the proportion of oil palm (OP) plantations (A1%) using an openly available dataset that covers OP distribution from 2001 to 2016 across Malaysia and Indonesia (Xu et al., 2020). To estimate the proportion of OP, we calculated the frequency of oil palm pixels in each 500 m × 500 m window.

(3) After accounting for the area of OP, the remaining forested area in each grid was further categorized into the evergreen broadleaf forest (EBF) (A2%) and other forest types (A3%) (i.e., deciduous broadleaf forest, coniferous forest, mixed forest, etc.), based on the ratio of EBF to the total forested area provided by MODIS Land Cover Type Product (MCD12Q1 v6.1) (Sulla-Menashe and Friedl, 2018).

(4) For the non-forested fraction of each grid cell, we used the latest version of the Land-use harmonization datasets (LUH2) dataset (Hurtt et al., 2020) to estimate the percentage of cropland (CRO) (B1%) and other non-forest land uses (B2%) (i.e., pasture, grass, etc.). "

[Figure]

**Figure 1: Workflow of the study. Steps (1) to (4) outline the processes for harmonizing multiple land cover datasets. Steps (5) to (6) show the establishment and interpretation of the LAI prediction machine learning model and the process of scenario simulations.**

**R2C6:** · Descriptions of scenario simulations are unclear. When introducing scenario simulation schemes, it is imperative to explicitly detail the calculation methods for assessing the impact of each factor, which would greatly enhance the manuscript's credibility and reliability.

**Response:** We apologize for the confusion. In the updated version, we explicitly included equations detailing the calculation process and improved the clarity of our statements for this section as follows:

"To quantify and compare the impacts of specific LUCC processes, climate change, and elevated $CO_2$ concentrations on vegetation greenness changes, we adopted the scenario simulation framework from several factorial attribution analyses (Sitch et al., 2015). Specifically, we first estimated the LAI trend under five hypothetical scenarios (S1 to S5) using the established XGBoost model. The equations are as below,

$$LAI_{i,t,Sn} = XGBoost(CO2_{i,t,Sn}, CLI_{i,t,Sn}, f\_EBF_{i,t,Sn}, f\_CRO_{i,t,Sn}, f\_OP_{i,t,Sn}, f\_Other_{i,t,Sn}) \quad (3)$$

$$\beta LAI_{i,t,Sn} = slope(LAI_{i,t,Sn}) \quad (4)$$

Where, $LAI_{i,t,Sn}$ represents the simulated LAI for the i[th] grid at year of t under scenario Sn and $\beta LAI_{i,t,Sn}$ indicates the LAI trend. The *XGBoost* stands for the established model for LAI prediction using $CO_2$ concentration, climate variable (CLI),

and land cover types such as fraction of evergreen broadleaf forest (f_EBF), cropland (f_CRO), oil palm (f_OP), other land uses (f_Other) (see Method 2.4).

For different scenarios, we adjusted the input variables according to specific assumptions to progressively incorporated different factors. For S1, we assumed only $CO_2$ concentration varies from 2001 to 2016, while climate and land uses variables (i.e., CLI, f_EBF, f_CRO, f_OP and f_Other) remained constant at their values in 2001. For S2, $CO_2$ and climate change over time, with land uses remaining unchanged since 2001. S3 to S5 sequentially considered different land use processes. S3 involved changes from EBF to CRO using time-varying $CO_2$, climate, and CRO area, while keeping OP and other land use types constant post-2001; S4 included conversions from EBF to both CRO and OP using time-varying $CO_2$, climate, CRO and OP areas, while other land uses unchanged since 2001; S5 encompassed all LUCC changes, with all variables including $CO_2$, climate, and all types of LUCC varying over time.

We then quantified the impacts of each factor on vegetation greening based on differences in LAI trends between scenarios,

$$Driver_n = \delta LAI\ trend = \beta LAI_{i,t,Sn} - \beta LAI_{i,t,Sn-1} \qquad (5)$$

Here, $Driver_n$ measures the impact of the $n^{th}$ driver (ranging from $CO_2$, climate change, CRO expansion, OP expansion, to Other LUCC) on LAI trends. Notably, $Driver_1$ quantifies the impact of $CO_2$, equal to $\beta LAI_{i,t,S1}$."

Minor comments

**R2C7:** 1. The discussion mentions, "It is also important to note that our estimation of CRO or OP expansion was based on the assumption that the increased areas of CRO or OP since 2001 came from EBF." Such a crucial assumption should be stated in the methodology section.

**Response:** Thank you for the suggestion. We have moved this statement to the methodology section Line 220 to Line 222.

**R2C8:** 2. On the basis of Figure 5, it would be beneficial to add the spatial distribution of dominant factors for each pixel. This enhancement would more clearly reveal whether the LAI trend for each pixel is positive or negative and which factors primarily drive these changes.

**Response:** Following the suggestion, we included the spatial distribution of dominant factors for each pixel by comparing the impacts of factors on the LAI trend (Fig. S6). Consistent with Figure 5, we found that the effect of $CO_2$ fertilization

dominated the increase in LAI in most areas, accounting for 62.10% of the study area. Conversely, CRO expansion was a dominant driver for greenness decline in many regions, accounting for 26.33% of the study area.

We have added the figure below in the supplementary file and included the following statement in the Result section, from Line xx to Line xx: "From a spatial perspective, we found that elevated $CO_2$ dominated the increase in LAI in most areas, accounting for 62.10% of the study area, while CRO expansion was the primary driver in LAI decrease in other regions (26.33%), especially coastal areas (Fig. S6)".

[Figure]

**Figure S6: Spatial pattern of dominant drivers of trend in LAI (a), and the percentage of the study area dominated by each diver (b). The drivers include elevated $CO_2$ ($CO_2$), climate change (CLI), crop expansion (CRO), oil palm expansion (OP) and other land use changes (Other). A prefix '+' of the drivers indicates a positive impact on LAI trends, whereas '−' indicates a negative impact.**

**R2C9:** 3. It would be preferable to represent Figure S2 as a scatter density plot (like Figure 4c,d) to facilitate the observation of changes in SHAP values with features, and to prevent potential misinterpretation arising from the clustering of scatter points.

**Response:** We agree that representing Figure S2 as a density plot will help avoid misinterpretation. Accordingly, we have revised this figure in the updated manuscript, as shown below.

[Figure]

**Figure S2: The density plots show the changes in SHAP values of each factor on LAI with corresponding factor variations. The abbreviations for each factor are available in Table S3.**

**R2C10:** 4. There is an error in Equation (2) that needs to be corrected.

**Response:** Thanks for pointing this out. We have corrected the Equation (2).

$$\phi_i, j(x) = \sum_{S \subseteq N\{i\}} \frac{|S|!\,(|N| - |S| - 2)!}{|N|!} [f(S \cup \{i, j\}) - f(S \cup \{i\}) - f(S \cup \{j\}) + f(S)] \quad (2)$$

**R2C11:** 5. Figure 4c,d depicts the coupling effects of f_EBF with f_OP and f_CRO rather than the interaction effects mentioned in the caption, making it seem

indistinguishable from Figure S2a. It is recommended to add SHAP dependence plots illustrating the interaction effects for a more in-depth analysis.

**Response:** Following the suggestion from the reviewer, we added SHAP interaction plots (Fig. 4e-f) on top of the SHAP dependence (coupling) plots (Fig. 4c-d) in the main text, to distinguish the figure from Fig. S2. We have also revised Fig. S2 to highlight the difference. We ensure that the caption provided for figure 4 is correct to avoid confusions. However, we refrained from overinterpreting the interaction plots (such as the interactions between f_EBF, f_OP and f_CRO), as by nature, we suspect that these three factors are likely dependent on each other (e.g., less f_EBF, more f_OP), not necessarily interact with each other (i.e., independent variables) in meaningful way.

[Figure]

**Figure 5: The impact of factors on LAI. (a) Bee swarm plots show the SHAP values of each factor on LAI for each sample. The SHAP value indicates the magnitude and direction of the impact on LAI (see Methods). Each dot represents an individual sample, with the color indicating the relative values of the specific factor. (b) The bar plot of the mean absolute SHAP**

values of each factor for LAI. (c) The interaction of f_OP and f_EBF, and (d) the interaction of f_CRO and f_EBF on LAI. The abbreviations for each factor are available in Table S3.

**R2C12:** 6. Previous studies have highlighted discrepancies between the cropland area changes provided by LUH2 and actual conditions in China and the United States. It is worth investigating whether a similar discrepancy exists in Southeast Asia. Meanwhile, the spatial resolution of the LUH2 dataset is too coarse for the purposes of this study.

**Response:** Regarding the accuracy of LUH2 in Southeast Asia, Mao et al. (2023) conducted a comparative analysis of LUH2 and eight other land-use products against a constructed land-use product for Southeast Asia. They compared several datasets, including remote sensing datasets like the MODIS Land Cover dataset (MCD), ESA CCI land cover maps (CCI), GLC_FCS30 (GLC), Copernicus Global Land Service Land Cover product (CGLS), and GlobeLand30 (GL), along with datasets from FAO, HYDE, and SAGE (Mao et al., 2023). Their analysis found that the cropland area estimates for this region were most closely aligned with those from LUH2, with a correlation coefficient ($r$) of 0.98 (Mao et al., 2023; Figure R1). This consistency indicates that LUH2 provides reliable cropland data for Southeast Asia.

[Figure]

**Figure. R1. Taylor diagrams comparing cropland area estimates with (a) LUH2, (b) HYDE, (c) SAGE (d) MCD, (e) GL, (f) CCI, (g) CGLS, (h) GLC, and (i) FAO data for subtropical East Asia and Southeast Asia. (Source: Mao et al., 2023)**

Mao F, Li X, Zhou G, et al. Land use and cover in subtropical East Asia and Southeast Asia from 1700 to 2018[J]. Global and Planetary Change, 2023, 226: 104157.

To examine the impact of spatial resolution on our results, we conducted a parallel analysis using spatial resolution of both 500m and 0.25 degree. We found our findings were robust to variations in spatial resolution, though the resolution of LUH2 is coarse. Specifically, we found the land use change and their impacts on the greening trends in our study area remain consistent across both 0.25-degree (Fig. S2, Fig. S6) and 500-meter (Fig. 2, Fig. 6) grid resolutions.

[Figure]

**Figure S2: Land use composition and its changes from 2001 to 2016 in the study area, analyzed at a 0.25-degree resolution.**

[Figure]

**Figure 2: Land use composition and its changes from 2001 to 2016 in the study area, analyzed at a 500-m resolution.**

[Figure]

**Figure S6: The spatial distribution of the pixel-wise impacts of each process on the greening trends, analyzed at a 0.25-degree resolution.**

[Figure]

**Figure 6: The spatial distribution of the pixel-wise impacts of each process on the greening trends, analyzed at 500-m resolution.**